# Ranitidine Alleviates Anxiety-like Behaviors and Improves the Density of Pyramidal Neurons upon Deactivation of Microglia in the CA3 Region of the Hippocampus in a Cysteamine HCl-Induced Mouse Model of Gastrointestinal Disorder

**DOI:** 10.3390/brainsci13020266

**Published:** 2023-02-04

**Authors:** Divya Bharathi Selvaraj, Jemi Feiona Vergil Andrews, Muthuswamy Anusuyadevi, Mahesh Kandasamy

**Affiliations:** 1Laboratory of Stem Cells and Neuroregeneration, Department of Animal Science, School of Life Sciences, Bharathidasan University, Tiruchirappalli 620024, India; 2Molecular Neuro-Gerontology Laboratory, Department of Biochemistry, School of Life Sciences, Bharathidasan University, Tiruchirappalli 620024, India; 3Faculty Recharge Programme, University Grants Commission (UGC-FRP), New Delhi 110002, India

**Keywords:** anxiety, cysteamine HCl, histamine, ranitidine, elevated plus maze, microglia

## Abstract

Elevated levels of histamine cause over-secretion of gastric hydrochloric acid (HCl), leading to gastrointestinal (GI) disorders and anxiety. Ranitidine is an antihistamine drug widely used in the management of GI disorders, as it works by blocking the histamine−2 receptors in parietal cells, thereby reducing the production of HCl in the stomach. While some reports indicate the neuroprotective effects of ranitidine, its role against GI disorder-related anxiety remains unclear. Therefore, we investigated the effect of ranitidine against anxiety-related behaviors in association with changes in neuronal density in the hippocampal cornu ammonis (CA)–3 region of cysteamine hydrochloride-induced mouse model of GI disorder. Results obtained from the open field test (OFT), light and dark box test (LDBT), and elevated plus maze (EPM) test revealed that ranitidine treatment reduces anxiety-like behaviors in experimental animals. Nissl staining and immunohistochemical assessment of ionized calcium-binding adapter molecule (Iba)-1 positive microglia in cryosectioned brains indicated enhanced density of pyramidal neurons and reduced activation of microglia in the hippocampal CA–3 region of brains of ranitidine-treated experimental mice. Therefore, this study suggests that ranitidine mediates anxiolytic effects, which can be translated to establish a pharmacological regime to ameliorate anxiety-related symptoms in humans.

## 1. Introduction

Gastrointestinal (GI) disorders and anxiety are widespread and raising clinical concerns worldwide [1,2]. The clinical characteristics of GI disorders are complex, and often known to be associated with anxiety-related symptoms [3,4,5]. The etiology and overlapping pathomechanisms responsible for GI disorders and anxiety remain largely unknown. Histamine is an important neuromodulator, recognized to be involved in homeostasis, allergies, immune functions, and neuroplasticity [6,7]. In the GI tract, histamine is released by the enterochromaffin-like cells of the gastric mucosa and it binds to the histamine-2 receptors of parietal cells which stimulates the secretion of hydrochloric acid (HCl) [8]. In addition, mast cells and basophils of the immune system are key sources of histamine in circulation [9]. In the brain, histaminergic neurons and microglia produce histamine, which acts as a neurotransmitter responsible for various neurophysiological functions [10]. While the physiological level of histamine is important for the functions of the GI tract, its elevated level appears to induce over-secretion of gastric HCl, leading to peptic ulcers [11,12]. The basal levels of histamine in the brain have been identified to play key roles in circadian rhythm, mood, and cognitive functions, whereas its elevated level causes pathogenic activation of microglial cells responsible for neuroinflammation in mood disorders and various neurocognitive impairments [6,13,14] (Figure 1). Therefore, antihistamine medications that are used to mitigate gastritis, allergies, and inflammation provide considerable relief against many neurological illnesses, psychotic episodes, and mood disorders, including anxiety [15,16,17]. However, the effects of antihistamines on the outcome of anxiety-related behavioral symptoms related to GI disorders have not been explored. The hippocampus is a crucial portion of the limbic system in the brain, which plays an important role in pattern separation, memory, and mood [18,19,20,21,22]. Mounting evidence suggests that the cornu ammonis (CA)-1 region of the hippocampus plays a vital role in fear conditioning, while abnormalities in the hippocampal CA-3 region appear to be involved in anxiety [23,24,25]. Considering the above facts, an increased level of histamine in the circulation or brain could be proposed as a pathogenic neuroimmune determinant and negative influencer of neuroplasticity in the CA3 region of the hippocampus, accountable for anxiety-related symptoms [25].

Many antihistamine drugs act as antagonists of histamine-1 receptor and are used against allergic reactions, while ranitidine is a potent blocker of histamine-2 receptors that has been widely used to treat various GI disorders including peptic ulcers [26,27]. Though the anti-gastric ulcerative effect of ranitidine is well established, its effect on brain function and behavioral outcomes remains largely undetermined. A growing body of experimental data supports the idea that ranitidine acts as an effective antidepressant. A new line of scientific evidence indicates that enhanced expressions of histamine-2 receptors in vulnerable subjects increase susceptibility to anxiety and depression [15,28,29]. Therefore, the pharmacological blockade of histamine-2 receptors is highly beneficial against many inflammatory disorders as well as mood disorders, including anxiety [30,31,32]. Besides, recent reports indicate that ranitidine treatment facilities analgesic, antioxidant, and cytoprotective effects in different organs, including the brain [33]. Moreover, ranitidine has been reported to modulate the immune cells thereby, reducing inflammation [34].

However, reports on the possible effect of ranitidine treatment on the regulation of neuroimmune function and anxiety are highly limited. Considering the aforementioned facts, it can be posited that ranitidine might reduce neuroinflammation as it can suppress the activation of microglia in the brain noticed during the disease conditions. Unexpectedly, few reports highlight the caution that ranitidine treatment induces some unprecedented adverse effects on the brain [35]. Thus, this study intended to investigate the effect of ranitidine against anxiety-related behaviors in a cysteamine HCl-induced mouse model of GI disorder in association with the difference in the number of pyramidal neurons and microglia in the CA3 region of the hippocampus of the brain.

## 2. Materials and Methodology

### 2.1. Experimental Animals

Three to four months old BALB/c mice were procured from the Biogen Laboratory, Bangalore, India, and maintained in the animal house facility of Bharathidasan University. The mice were kept in an air-conditioned room at 20–22 °C with a 12-h light/dark cycle. The experimental mice were freely allowed to access standard animal feed and water. A total of 24 experimental mice were randomly divided into four groups, namely, (1) control group (N = 6), (2) cysteamine HCl group (N = 6), (3) ranitidine group (N = 6), and (4) cysteamine HCl + ranitidine group (N = 6). While mice in group 1 received normal tap water, mice in group 2 and group 4 received an intraperitoneal injection of cysteamine HCl (Sigma Aldrich, St. Louis, Missouri, USA) (60 milligrams (mg)/kilogram (Kg) body weight (BW)) for three alternate days to induce GI disorder. While a minimal dose of ranitidine can be safe and effective, in this candidate approach, the dose of ranitidine has been determined based on previous studies [36]. Thus, animals in groups 3 and 4 were orally administered ranitidine (Cadila Pharmaceuticals Limited, India) (30 mg/ Kg BW) in drinking water for 14 consecutive days. After the treatment, the experimental mice underwent behavioral experiments such as the open field test (OFT), light and dark box test (LDBT), and elevated plus maze (EPM) test. The behavioral room was equipped with a proper light setting. A video camera was placed above the center of the behavioral apparatuses. The camera was connected to a semi-automated computer system equipped with the SMART 3.0 video tracking system (Panlab, Harvard apparatus, Spain) through which digital tracking of all the animal behavioral experiments was captured (Figure 2). After the behavioral experiments, the animals were perfused, and brain tissues were processed for histology and immunostainings as earlier described by Kandasamy and colleagues [19,37].

The experimental protocol involving animals was approved (Ref No. BDU/IAEC/P10/2019 dated 30.11.2019) by the Institutional Animal Ethics Committee (IAEC), Bharathidasan University under the regulation of the Committee for the Purpose of Control and Supervision of Experiments on Animals (CPCSEA), Government of India.

### 2.2. Open Field Test

The OFT was conducted to assess the locomotive and anxiety-like behaviors in the experimental mice. A standard wooden arena (120 cm × 120 cm) consisting of 16 grids, sized 30 cm each, was used for the OFT. The arena was digitally subdivided into an outer zone (red color) and an inner zone (dark green) using SMART 3.0. Each animal was gently released into the center of the arena and allowed to freely explore the entire field without any interruption. Three trials were conducted for three consecutive days with a duration of 5 minutes (min) each. The locomotive path, total number of grids crossed, distance traveled, and time spent in the outer zone, and inner zone were recorded and calculated for each animal using SMART 3.0. After the OFT, the animal was returned to its home cage. At the end of each trial, the open field arena was cleaned with 70% ethanol [5,19,20].

### 2.3. Light and Dark Box Test

The LDBT was used to determine the photic-related preferences and unconditioned avoidance or preference responses in the experimental mice. A standard black, wooden, rectangular box consisting of close/dark, and open/light compartments connected via a slit was used for the LDBT. Using SMART 3.0, two zones were digitally inserted to designate the dark and light compartments. An area outlined with blue color represented the covered untrackable dark area, and a red color denoted the trackable, open compartment exposed to light. Each mouse was placed in the middle of the apparatus and allowed to explore both dark and bright areas without any disturbance. Three trials were conducted for three consecutive days with a duration of 5 min per trial. After the experiments, each mouse was gently returned to the home cage. The entire apparatus was wiped with 70% ethanol and dried. The total time spent in the light and dark compartments by experimental animals was estimated from trajectories recorded using SMART 3.0 [5,20].

### 2.4. Elevated Plus Maze

The EPM was used to analyze the anxiety in the experimental mice. The EPM apparatus consisted of 4 walking arms connected in the middle zone. The two arms were protected by 30 cm high sidewalls, and the two open arms were kept unprotected. The whole EPM setup was 90 cm elevated from ground level. Using SMART 3.0, all four arms were digitally designated as four different areas with color marks. The two closed arms were marked with blue and brown colors, and the two open arms were marked with red and violet colors, respectively. Each animal was gently placed in the middle zone and allowed to explore all four arms for 5 min. Three trials were carried out for three consecutive days. The time spent by each animal in the open and closed arms was calculated using SMART 3.0. Upon the completion of the trial, the animal was placed back into the home cage. The whole EPM was cleaned with 70% ethanol and allowed to evaporate after each trial [5,20].

### 2.5. Perfusion of Animals and Brain Tissue Processing

The experimental mice were deeply anesthetized and transcardially perfused with 0.9% sterile saline followed by 4% paraformaldehyde (PFA) (Himedia, India). The brains were dissected from the skulls of animals and soaked in 4% PFA for 24 hours (h). Then the brains were transferred into separate sterile tubes containing 30% sucrose (SRL, India) and maintained at 4 °C. After a week, each brain was molded in a solution of optimal cutting temperature compound (OCT) (Sigma Aldrich, St. Louis, MO, USA) and placed on the tissue holder of a sliding microtome (Weswox, India) and the brain was frozen using dry ice. Each brain was cut into 30 µm serial sagittal sections and systematically collected in 12 sterile tubes containing a mixer of cryoprotectant solution made up of glycerol (Merck, Germany), ethylene glycol (Himedia, India), and a phosphate buffer in a 1:1:2 ratio and stored at −20 °C. The brain sections, 1 out of 12 tubes (360 µm apart) were used for histological and immunohistochemical assessments [19,37].

### 2.6. Nissl Staining

The cryosections were mounted on (3-Aminopropyl) triethoxysilane (APTES) (SRL, India)-coated microscopic glass slides (Borosil, India) and air-dried overnight. Then, 100 mg of cresyl violet acetate (SRL, India) was dissolved in 100 mL of distilled water together with 250 µL of glacial acetic acid (SRL, India) using a magnetic stirrer (Remi, India) at 60 °C. After bringing it down to room temperature, the cresyl violet solution was then filtered using Whatman filter paper. First, the slides containing brain sections were immersed in an alcohol–chloroform (1:1) solution for 3 min, and the sections were rehydrated through 100%, 95% ethanol, and double distilled water for 3 min. Then, the sections were stained with cresyl violet solution for 30 min. Further, the brain sections were rinsed with double distilled water followed by 100%, and 95% of ethanol treatment for 3 min, followed by xylene (Merck, Germany). The sections were air-dried and mounted with dibutyl phthalate polystyrene xylene (DPX) (Merck, Germany). The pyramidal cells were counted in the hippocampus using a light microscope (Leica Microsystems, Germany) connected to a computer with the ImageJ plugin with a cell counter as described by Yesudas et al. [19].

### 2.7. Immunohistochemistry of Microglia

The free-floating brain sections were washed thrice with Tris-buffered saline (TBS) for 10 min in a 12-well plate (Tarson, India), then the sections were placed in a sodium citrate buffer at 65 °C for 90 min for antigen retrieval. After incubation, the sections were washed thrice with TBS for 10 min. Then, the sections were blocked with 3% bovine serum albumin (BSA) (Himedia, India), for an hour before being treated with the primary antibody. The brain sections were transferred to a solution containing a rabbit α ionized calcium-binding adapter molecule (Iba)-1 antibody with the dilution of 1:250 (Cell Signaling Technology, Danvers, MA, USA) and incubated at 4 °C for 48 h. After incubation with the primary antibody, the sections were washed with TBS for 10 min. Further, brain sections were incubated with a goat α rabbit DyLight^TM^ 594 secondary antibody with the dilution of 1:500 (Novus Biologicals, Colorado, USA), which was added and kept at 4 °C for 24 h. The next day, the sections were washed twice with TBS for 10 min. The sections were incubated with 0.1 mg/mL of 4′,6-diamidino-2-phenylindole (DAPI) (Himedia, India) for 5 min, and washed again with TBS for 10 min. After that, the sections were mounted with ProLong^TM^ Glass antifade mountant (Thermo Fisher Scientific, Waltham, MA, USA) and dried overnight. The slides were blind-coded, and the brain sections were analyzed and photographed using an epifluorescence microscope (Leica Microsystems, Germany). The total number of Iba-1 immune-positive cells was counted in 5 different non-overlapping photomicrographs in the CA3 of the hippocampus using the ImageJ plugin with a cell counter. In addition, 50 cells in the brain of each animal were assessed for the morphology of Iba-1 positive cells based on the ramified, hypertrophic, and amoeboid characteristics to obtain the percentage of different types of microglia cells in the CA3 of the hippocampus.

### 2.8. Statistical Analyses

The values have been represented as mean ± standard deviation (SD). Results from the behavioral, histological, and immunohistochemical assessments were assessed by one-way analysis of variance (ANOVA) followed by Tukey’s post hoc test for multiple comparisons. Graph Pad Prism was used to assess all the statistical analyses. The level of statistical significance was considered at *p* < 0.05 unless otherwise indicated.

## 3. Results

### 3.1. Ranitidine Treatment Increased the Locomotory Exploration but Minimized Anxiety-Related Behaviours of Mice in the Cysteamine HCl-Treated Group in OFT

In the assessment of locomotion-based exploratory behavior in the OFT, the total number of grids crossed by the cysteamine HCl-treated mice was found to be less when compared to the control, ranitidine alone, and cysteamine HCl + ranitidine-treated groups. Ranitidine-treated animals crossed significantly more grids than mice in the control, cysteamine HCl treated group, and cysteamine HCl + ranitidine-treated groups. Mice in the cysteamine HCl + ranitidine-treated groups also crossed more grids than the mice in the cysteamine HCl-treated group (CTR = 66 ± 40; CYS = 37 ± 15; RNT = 101 ± 32; CYS + RNT = 73 ± 24) (Figure 3).

In addition, the total distance traveled by the cysteamine HCl-treated animals was reduced than mice in the control, ranitidine alone, and cysteamine HCl + ranitidine-treated groups. The total distance traveled by mice in the ranitidine alone treated group was found to be increased than that of mice in the control, cysteamine HCl and cysteamine HCl + ranitidine-treated groups. The distance traveled by mice in the cysteamine HCl + ranitidine-treated group was significantly increased than the cysteamine HCl-treated group (CTR = 65 ± 27; CYS = 40 ± 17; RNT = 86 ± 20; CYS + RNT = 60 ± 18) (Figure 3)**.**

With reference to preference-based anxiety behaviors in the OFT, experimental mice in the cysteamine HCl-treated group spent more time in the outer zone when compared to mice in the control, ranitidine alone, and cysteamine HCl + ranitidine-treated groups (CTR = 262 ± 32; CYS = 288 ± 67; RNT = 224 ± 30; CYS + RNT = 259 ± 21). As a result, the time explored in the inner zone by cysteamine HCl-treated animals was considerably reduced when compared to mice in the control, ranitidine alone, and cysteamine HCl + ranitidine-treated groups (Figure 3)**.** The experimental mice in the ranitidine alone-treated group explored for markedly more time in the inner zone compared to mice in the control, cysteamine HCl, and cysteamine HCl + ranitidine-treated groups. Moreover, the time spent in the inner zone by cysteamine HCl + ranitidine-treated mice was also significantly increased over mice treated with cysteamine HCl alone (CTR = 38 ± 32; CYS = 12 ± 7; RNT = 76 ± 30; CYS + RNT = 41 ± 21) (Figure 3). Overall, the results obtained from the OFT indicate that ranitidine treatment abolishes the cysteamine HCl-induced abnormalities in locomotive exploratory and anxiety-related behavioral deficits in experimental animals.

### 3.2. Ranitidine Treatment Reduced Preference-Based and Unconditioned Anxiety-like Behaviors in Cysteamine HCl-Treated Mice in the LDBT

During the LDBT, the experimental mice in the cysteamine HCl-treated group stayed for a longer time in the dark compartment when compared to other groups (CTR = 200 ± 41; CYS = 242 ± 34; RNT = 138 ± 59; CYS + RNT = 203 ± 33). Hence, the time spent in the light compartment by mice in the cysteamine HCl group was significantly reduced when compared to the control, ranitidine alone, and cysteamine HCl + ranitidine-treated groups. The time spent in the dark compartment by the ranitidine alone-treated mice was considerably reduced, as they explored for a longer time in the light compartment than the control, cysteamine alone, and cysteamine HCl + ranitidine-treated groups. Notably, the time spent in the dark compartment by mice in the cysteamine HCl + ranitidine-treated group was also considerably reduced than mice in the cysteamine HCl alone treated group (CTR = 100 ± 41; CYS = 58 ± 34; RNT = 162 ± 59; CYS + RNT = 97 ± 33). The results obtained from the LDBT suggest that ranitidine treatment decreases cysteamine HCl-induced anxiety-like symptoms in experimental animals (Figure 4)**.**

### 3.3. Ranitidine Treatment Reduced Acrophobia-Related Anxiety-like Behaviors in Cysteamine HCl-Treated Mice in the EPM

Upon exposure to the EPM, the experimental mice in the cysteamine HCl-treated group were found to spent more time in the protected closed arms of the EPM, while their activities to explore the open arms were drastically decreased when compared to the control, ranitidine alone, and cysteamine HCl + ranitidine-treated groups (CTR = 288 ± 18; CYS = 298 ± 4; RNT = 268 ± 26; CYS + RNT = 287 ± 10). In contrast, the experimental mice treated with ranitidine alone showed a higher tendency to explore the open arms over the control, cysteamine HCl, and cysteamine HCl + ranitidine-treated groups. Moreover, the time spent in the open arms by mice in the cysteamine HCl + ranitidine-treated group was also considerably increased over the mice treated with cysteamine HCl (CTR = 12 ± 18; CYS = 2 ± 4; RNT = 32 ± 26; CYS + RNT = 13 ± 10). Hence, the results obtained from the EPM corroborate that ranitidine treatment mitigates cysteamine HCl-mediated anxiety in experimental animals (Figure 5).

### 3.4. Ranitidine Treatment Increased the Neuronal Density in the CA3 Regions of the Hippocampi in the Brains of Cysteamine HCl-Treated Mice

Recent experimental proof suggests that abnormal changes in pyramidal neurons in the CA3 region of the hippocampus contribute to anxiety-related behaviors. Thus, the cresyl violet-stained brain sections prepared from experimental animals were examined for morphology and density of Nissl bodies in the CA3 region of the hippocampus. In the cysteamine HCl-treated group, there were omnipresent signs of the loss of cellular integrity and degenerative-like changes in the CA3 region of the hippocampus. The number of pyramidal neurons was significantly reduced in the CA3 region of the hippocampus in the brains of mice in the cysteamine HCl-treated group compared to the control, ranitidine alone, and cysteamine HCl + ranitidine-treated groups, whereas in the ranitidine alone-treated group, the number of pyramidal neurons was significantly increased in the CA3 region of the hippocampus of the brain compared to that of the control, cysteamine HCl, and cysteamine HCl + ranitidine-treated groups. Notably, mice in the cysteamine HCl + ranitidine-treated group also exhibited an increased number of pyramidal neurons in the hippocampal CA3 region compared to the cysteamine HCl-treated group (CTR = 91 ± 14; CYS = 57 ± 14; RNT = 146 ± 28; CYS + RNT = 109 ± 26) (Figure 6). Thus, the results suggest that ranitidine treatment provides neuroprotection and rejuvenation against cysteamine HCl-induced neuropathogenic modifications in the hippocampal CA3 region of the brain.

### 3.5. Ranitidine Treatment Attenuated Cysteamine HCl-Induced Activation of Microglia in the Hippocampal CA3 Region of the Brain

As microglial activation in the brain has increasingly been identified as a pathogenic determinant of anxiety and aberrant hippocampal plasticity, this study was extended to assess the changes in Iba-1-positive microglia in the CA3 region of the hippocampus in the brains of experimental animals. Surprisingly, the overall number of microglial cells, regardless of morphological changes, was found to be increased in the CA3 region of the hippocampus in the brains of the experimental mice in all the treatment groups when compared to the control. However, a decreased number of microglial cells was evident in the CA3 region of the hippocampus in the brains of the mice in ranitidine-treated groups when compared to cysteamine HCl-treated and cysteamine HCl + ranitidine-treated groups (CTR = 79 ± 14; CYS = 126 ± 17; RNT = 108 ± 9; CYS + RNT = 118 ± 29) (Figure 7). This led us to assess and compare the morphological differences of microglial cells with ramified, hypertrophic, and amoeboid-like features in the CA3 region of the hippocampus of the brains among experimental groups. While experimental mice in the cysteamine HCl-treated group showed a considerable raise in the proportion of activated microglia with hypertrophic and amoeboid-like characteristics, the percentage of activated microglial cells was found to be reduced in the CA3 region of the hippocampus in the brains of mice in the ranitidine alone treated group compared to the cysteamine HCl-treated and cysteamine HCl + ranitidine-treated groups. Eventually, a non-significant reduction in the percentage of activated microglial cells was also noticed in the CA3 region of the hippocampus in the brains of the cysteamine HCl + ranitidine-treated mice than the cysteamine HCl-treated group (% of ramified microglia: CTR = 81.5 ± 8; CYS = 45 ± 7; RNT = 61 ± 6; CYS + RNT = 49 ± 7) and (% of activated microglia: CTR = 18.5 ± 8; CYS = 55 ± 7; RNT = 39 ± 6; CYS + RNT = 51 ± 7). Considering the aforementioned facts, induced microglial activation upon exposure to cysteamine HCl appears to be attenuated by ranitidine-mediated effects in the brain during anxiety associated with GI disorder.

## 4. Discussion

The present study demonstrates that ranitidine increases locomotion-based exploratory behaviors and reduces anxiety in association with increased neuronal density and deactivation of microglial cells in the hippocampal CA3 region of a cysteamine HCl-induced mouse model with GI disorder. Recent scientific evidence unequivocally shows that the pathogenicity of GI disorders seriously affects the brain, thereby leading to anxiety-related symptoms. However, the overlapping pathomechanism between GI disorders and anxiety remains unknown. The overproduction of histamine in the stomach has been identified as a cause of GI disorders and can also lead to the development of anxiety. In the brain, histamine plays a crucial role in the regulation of neurotransmission of acetylcholine (ACh), serotonin, gamma-aminobutyric acid (GABA), and vasopressin [6]. Histamine has been identified to be important for the physiology of glial cells, thermoregulation, energy metabolism, and blood–brain barrier permeability [10]. Moreover, histamine has been known to be involved in the regulation of the HPA and HPG axes [38,39]. While aberrant levels of histamine-mediated alteration in the synthesis and downstream signaling of corticosteroid hormones and the reuptake process of serotonin by glial cells have increasingly been recognized in the development of stress and depression, the pathomechanism leading to anxiety caused by elevated histamine remains obscure [29].

A potential crosstalk between peripheral immune cells and brain-resident immune cells during various pathogenic processes appears to be linked to anxiety [40]. Notably, excess levels of histamine appear to induce anxiogenic effects through the activation of neuroimmune cells in the brain [41]. In general, the hippocampus of the brain plays a key role in various cognitive functions [18]. Recent association studies and animal experiments strongly indicates that alteration in hippocampal volume, distorted pyramidal neurons, and microglial activation in the CA3 region of the hippocampus can contribute to anxiety. Gene expression studies have revealed that microglia express all histamine receptors; thus, surplus circulation of histamine can stimulate microglial activation leading to the subsequent production of free radicals and proinflammatory factors, including tumor necrosis factor (TNF)-α and Interleukins (IL)-6 in the hippocampus [42]. Considering this fact, neuroinflammation resulting from activated microglia may lead to detrimental effects on hippocampal neurons in the brain [43]. Notably, many antihistamines have been known to exhibit antidepressant and anxiolytic properties. Among various antihistamine drugs developed, ranitidine has been widely used to treat GI disorders, as it effectively reduces the gastric secretion of HCl [44]. While the anti-inflammatory properties of ranitidine have been increasingly evident, previous research findings provided evidence that ranitidine counteracts pathogenicity resulting from traumatic brain injury [45]. Moreover, ranitidine treatment has been suggested to be effective in treating clinical symptoms of stress and depression. Though some reports indicated the anxiolytic effect of ranitidine, studies for its neurotherapeutic roles, and possible cellular mechanism in the brain through which it attenuates the clinical symptoms of anxiety, need to be established. Moreover, the effects of ranitidine against anxiety at the level of microglial activation and the density of pyramidal neurons in the hippocampus in the brain of subjects with GI disorders have not been addressed.

In this study, we used the OFT, LDBT, and EPM to examine the effects of ranitidine on the outcome of innate and GI disorder-associated anxiety-like behaviors in control and cysteamine HCl-treated animals, respectively. Results revealed that cysteamine HCl-treated animals exhibited enhanced anxiety-related behaviors. This data validates our previous findings that cysteamine HCl treatment can mediate ulcerative problems in the GI tract and induce oxidative stress and neuroinflammation in the brain leading to anxiety [5]. The neuroinflammation seen in various GI disorders has been known to induce pathogenic alteration in the hippocampus affecting the morphological features and neuroplasticity of the pyramidal neurons predominately in the CA3 region of the hippocampus, which could be associated with an increased level of histamine [46,47]. In addition, the infusion of exogenous histamine into the substantia nigra and striatum of experimental animals resulted in neurodegeneration associated with the activation of microglia [48]. Therefore, the prominent anxiety-related behavior noticed in cysteamine HCl-treated mice could be related to an increased level of histamine-mediated neuropathogenic changes in the brain. In contrast, ranitidine-treated mice exhibited less anxiety than the control or cysteamine HCl-treated animals. Despite its well-known anti-ulcerative effect in the GI tract, ranitidine has been known to mediate anti-inflammatory and cytoprotective effects [30,49,50]. In this study, the quantification of Nissl-stained neurons revealed an increase in the density of pyramidal neurons in the CA3 region of the hippocampus in the brain of ranitidine alone and cysteamine HCl + ranitidine-treated groups over that of cysteamine HCl-treated mice. Notably, Malagelada and coworkers demonstrated that ranitidine treatment provides neuroprotection in cultured rat brain cortical neurons largely due to reduced levels of Caspase-3-mediated apoptotic signaling [51]. Another study by Park et al. reported that ranitidine provides neuroprotection against rotenone-induced apoptosis, as it inhibits the phosphorylation of c-Jun N-terminal Kinase (JNK) and P38 in human dopaminergic SH-SY5Y cells [52]. Thus, the enhanced neuronal density noticed in the hippocampal CA3 region of the ranitidine-treated groups can presumably be related to the enhanced density of pyramidal neurons due to its possible anti-apoptotic signaling-based neuroprotective mechanism and or neurorestorative process in the brain. Tricyclic antidepressants and selective serotonin reuptake inhibitors are the most commonly used drugs for anxiety-related disorders, as they facilitate serotonergic functions. Choa et al. reported that ranitidine treatment and tiotidine in experimental animals increased the release of serotonin in the brain [53]. Therefore, it can be expected that the efficacy of ranitidine may be similar to that of other existing anxiolytic medications. However, further comparative-study and valid experiments on the effects of ranitidine in comparsion with widely used anxiolytic drugs are needed. Ranitidine has also been administered through an intravenous route. However, the intravenous administration of ranitidine might be associated with cardiac arrest and other undesirable adverse effects [54]. Therefore, oral intake of ranitidine can be an ultimate non-invasive method for therapeutic purposes, as it can directly or indirectly influence brain function.

Recently, ramified microglia have been known to exhibit neuroprotective functions in the hippocampus and contribute to synaptic plasticity [55,56], whereas activated microglial cells with hypertrophic and amoeboid features have been known to induce deleterious effects on neurons of the brain during neuropathogenic conditions [57,58]. Taken together, it can be hypothesized that ranitidine treatment mitigates cysteamine HCl-induced anxiety-related behaviors at the level of providing neuroprotection in association with the replenishment and modulation of microglial cells in the CA3 region of the hippocampus of the brain. However, the molecular mechanisms by which ranitidine treatment facilitates anxiolytic effects through neuroprotection or neuro rejuvenation in association with or independent of microglia remains less clear. Thus, future studies are essential to understand the dysregulation of the histaminergic system and mechanisms of ranitidine in health and disease with regard to the regulation of neuroplasticity responsible for cognition, mood, and emotion.

## 5. Conclusions

Ranitidine is a selective blocker of histamine-2 receptors, which effectively reduces acid secretion in the stomach. Results obtained from animal behavioral studies revealed that ranitidine mitigates anxiety-like behavior in an experimental animal model with GI disorder. The study provides further evidence that the anxiolytic effect of ranitidine could be associated with the enhanced density of pyramidal neurons and reduction of activated microglia in the CA3 region of the hippocampus in the brain of experimental animals. Considering the facts, ranitidine could facilitate neuroprotective and anti-inflammatory effects in the brain. The current study advocates that ranitidine can be considered a candidate drug in the treatment regime for anxiety-related symptoms. Moreover, ranitidine can be considered for implementation in the treatment of brain diseases resulting from neurodegeneration and neuroinflammation. However, further experimental validations are required for its role in the cell cycle regulation of neuronal and non-neuronal cells and apoptotic events in the brain. Moreover, possible adverse effects resulting from improper and prolonged intake of ranitidine cannot be completely excluded.

## Figures and Tables

**Figure 1 brainsci-13-00266-f001:**
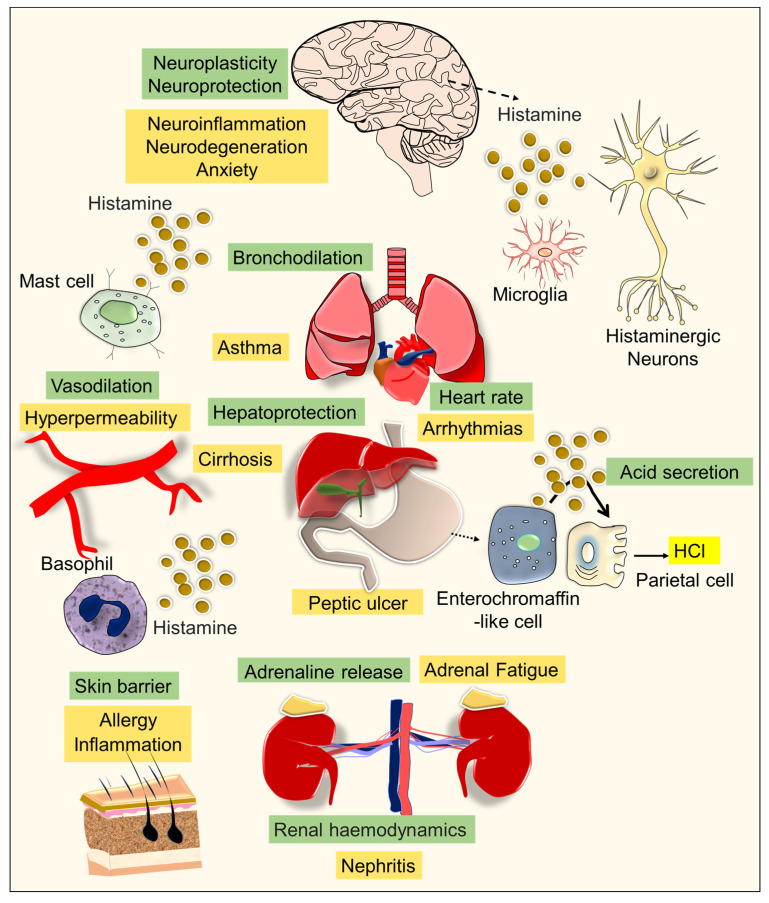
The figure depicts the role of histamine in normal and pathophysiological conditions. Histamine is an important neuromodulator that facilitates neuroplasticity and neuroprotection in the brain. Histamine regulates heart rate, bronchodilation, gastric acid secretion, maintains renal hemodynamics, and plays a vital role in hepatoprotection and adrenaline release. Histamine helps with vasodilation and provides a skin barrier (the role of histamine in the physiological function is indicated in the text in the olive-colored background). In certain diseases or exposure to allergens, overproduction of histamine levels leads to peptic ulcers, inflammation, asthma, arrhythmias, nephritis, liver cirrhosis, adrenal fatigue, hyperpermeability of blood vessels, neurodegeneration, and anxiety (the role of abnormal levels of histamine in the diseased conditions is highlighted in the text with the mustard-colored background).

**Figure 2 brainsci-13-00266-f002:**
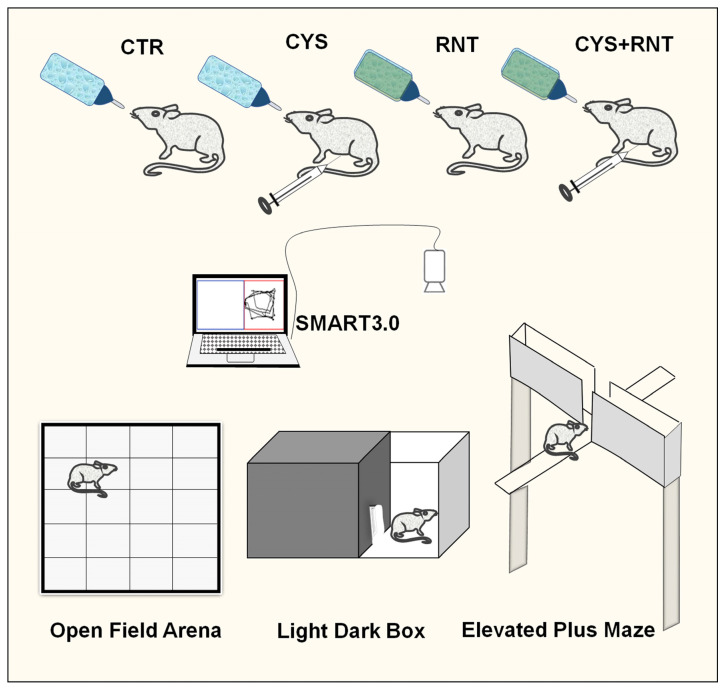
The figure represents the experimental grouping of animals and treatment in control (CTR), cysteamine HCl (CYS), ranitidine (RNT), and cysteamine HCl + ranitidine (CYS + RNT). The animals were subjected to various behavioral experiments such as the open field test, light and dark box test, and elevated plus maze test. The activities of animals during the behavioral experiments were tracked and recorded using a computer with SMART 3.0 software via a wall-mounted camera.

**Figure 3 brainsci-13-00266-f003:**
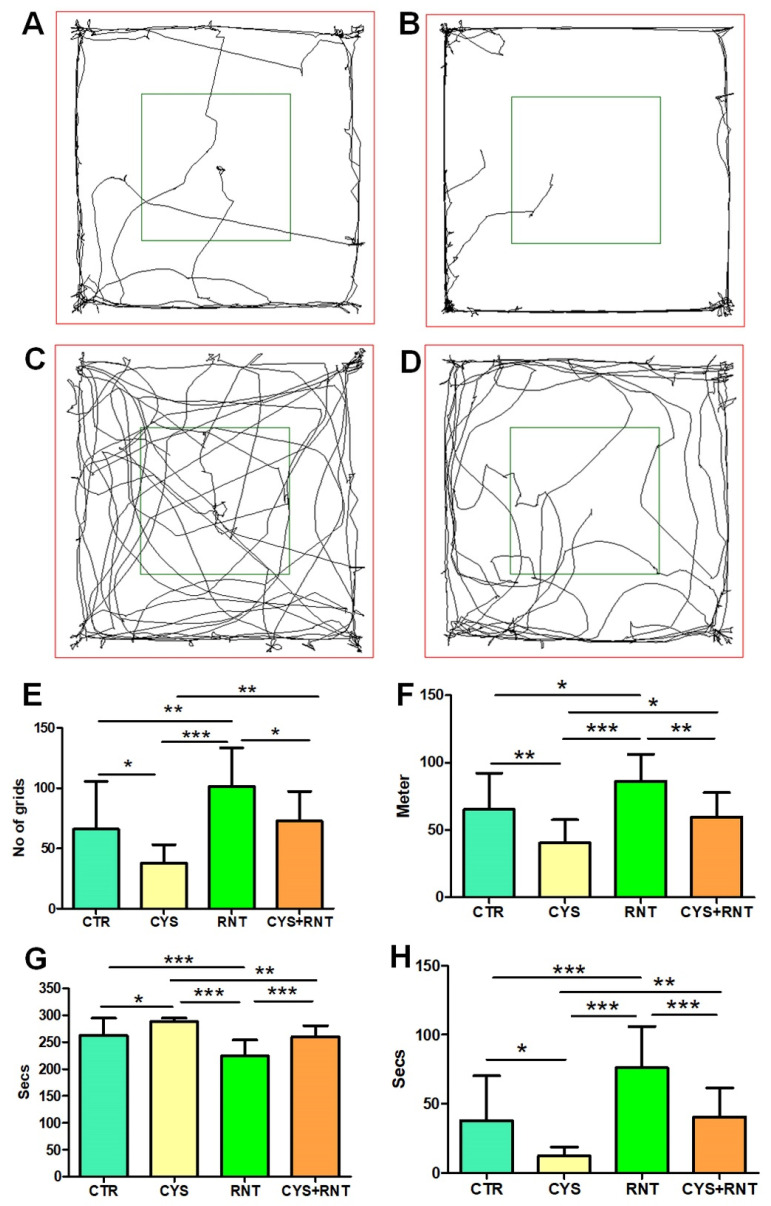
Ranitidine improved locomotory exploration and reduced anxiety behaviors in cysteamine HCl-treated animals in the open field test (OFT). The images represent video tracking of an animal from the (**A**) control (CTR), (**B**) cysteamine HCl (CYS), (**C**) ranitidine (RNT), and (**D**) cysteamine HCl + ranitidine (CYS + RNT)-treated groups during the OFT. The red square indicates the outer zone, and the green square indicates the inner zone of the test arena. The bar graph data indicates (**E**) the number of grids crossed, (**F**) the distance traveled, (**G**) time spent in the outer zone, and (**H**) the inner zone by mice in the control and treatment groups (* *p*-value ≤ 0.05, ** *p*-value ≤ 0.01 and *** *p*-value ≤ 0.001).

**Figure 4 brainsci-13-00266-f004:**
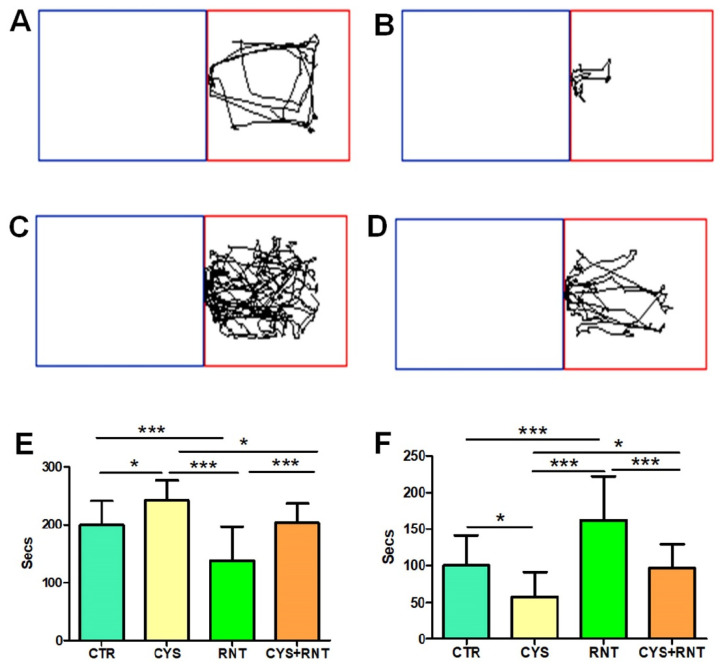
Ranitidine decreases preference-based anxiety-like behaviors in cysteamine HCl-treated mice during the light–dark box test (LDBT). The images represent video tracking in LDBT of an animal from the (**A**) control (CTR), (**B**) cysteamine HCl (CYS), (**C**) ranitidine (RNT), and (**D**) cysteamine HCl + ranitidine (CYS + RNT)-treated groups. The area outlined with blue represents the dark compartment, and the red color denotes the light compartment. The bar graph data indicates the time spent in (**E**) the dark and (**F**) the light compartments by mice in the control and treatment groups (* *p*-value ≤ 0.05 and *** *p*-value ≤ 0.001).

**Figure 5 brainsci-13-00266-f005:**
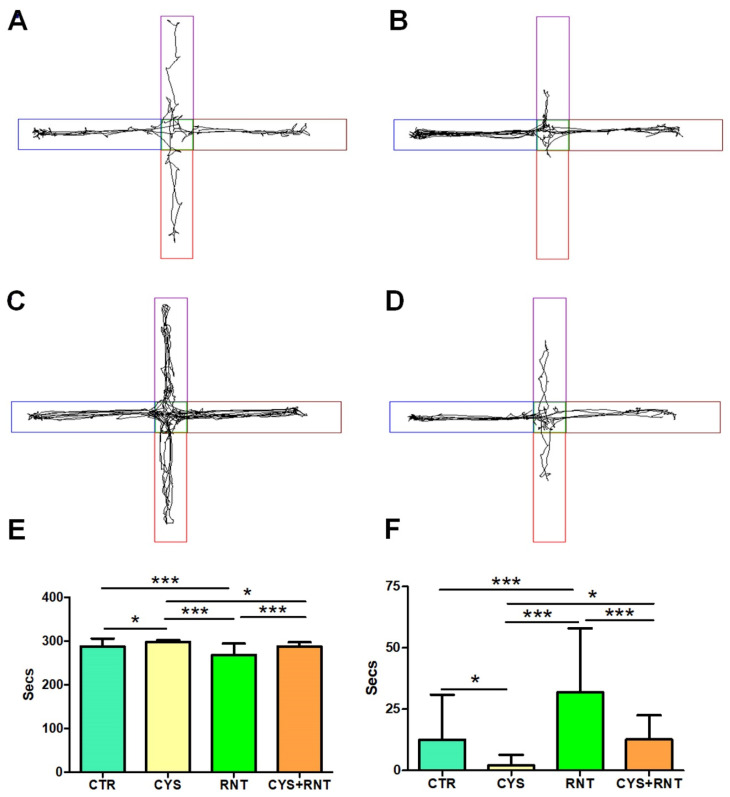
Ranitidine treatment exhibits anxiolytic behaviors in cysteamine HCl-treated mice in the elevated plus maze (EPM). The images represent video tracking in the EPM of an animal from the (**A**) control (CTR), (**B**) cysteamine HCl (CYS), (**C**) ranitidine (RNT), and (**D**) cysteamine HCl + ranitidine (CYS + RNT)-treated groups. Two closed arms were marked as blue and brown, and two open arms were marked as red and violet, respectively. The bar graph data indicates the time spent in (**E**) the closed arms and (**F**) the open arms by mice in the control and treatment groups (* *p*-value ≤ 0.05 and *** *p*-value ≤ 0.001).

**Figure 6 brainsci-13-00266-f006:**
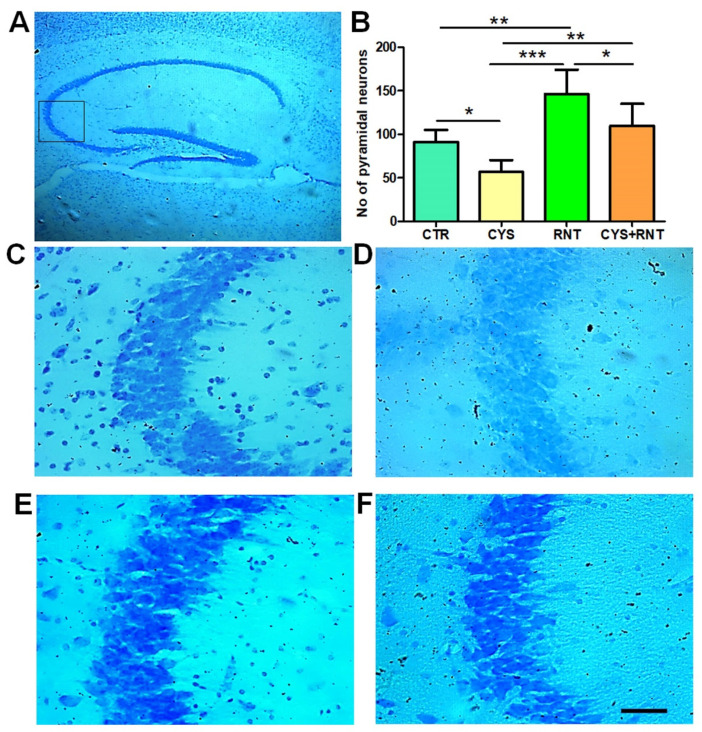
Ranitidine treatment increased neuronal density in the CA3 region of the hippocampus in cysteamine HCl-treated mice. The images represent the following: (**A**) the overall hippocampus with a black square indicating the CA3 region; (**B**) the bar graph data indicates the number of Nissl-stained cells in the hippocampal CA3 region of experimental mice followed by the representative enlarged microscopic images of Nissl staining of the hippocampal CA3 region of the brain from (**C**) control (CTR), (**D**) cysteamine HCl (CYS), (**E**) ranitidine (RNT), and (**F**) cysteamine HCl + ranitidine (CYS + RNT)-treated groups (* *p*-value ≤ 0.05, ** *p*-value ≤ 0.01 and *** *p*-value ≤ 0.001). Scale bar = 50 µm.

**Figure 7 brainsci-13-00266-f007:**
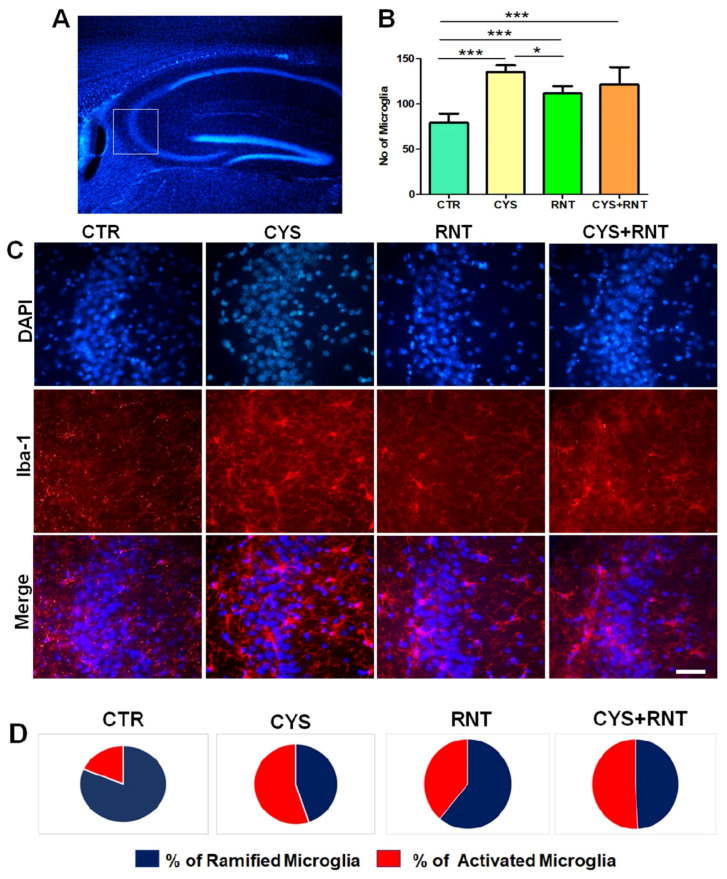
Ranitidine treatment decreased the number of Iba-1-positive cells in the CA3 region of the hippocampus in cysteamine HCl-treated mice. The images represent the following: (**A**) an overall DAPI-stained hippocampus with a white square indicating the CA3 region; (**B**) the bar graph data indicates the number of Iba-1-positive cells in the hippocampal CA3 region of experimental mice; (**C**) the fluorescence staining representative-enlarged microscopic images of DAPI, Iba-1, and overlay of the same in the hippocampal CA3 of the brain from control (CTR), cysteamine HCl (CYS), ranitidine (RNT), cysteamine HCl + ranitidine (CYS + RNT)-treated groups, Scale bar = 50 µm; and (**D**) the pie chart denotes the percentage of ramified and activated microglial cells in the hippocampal CA3 regions of the experimental mice (* *p*-value ≤ 0.05 and *** *p*-value ≤ 0.001).

## Data Availability

All data needed to evaluate the conclusions are present in the paper.

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
