# Peer review of "Ranitidine Alleviates Anxiety-like Behaviors and Improves the Density of Pyramidal Neurons upon Deactivation of Microglia in the CA3 Region of the Hippocampus in a Cysteamine HCl-Induced Mouse Model of Gastrointestinal Disorder"

_brainsci, 2023, doi:10.3390/brainsci13020266_

Round 1

Reviewer 1 Report

The current manuscript presents a quite interesting study on the anxiolytic mechanism of ranitidine. It is well done and presents relevant results. Nevertheless, some improvements should be made:

- The abstract should not have abbreviations, hence they should be removed; also the abstract lacks a final sentence, that corresponds to the study’s general conclusion(s);

- A schematic representation should be done in the introduction section, regarding the pathophysiological aspects of the connection between histamine and its effects in the gastrointestinal tract and brain (easier for the reader to visualize);

- A representation of the different pharmacodynamic in vivo tests, such as the open field test, light and dark box test and elevated plus maze, should be done in the form a figure, for better reader comprehension;

- The figures should be provided in higher resolution; also they should not have a title imbedded, such as “Fig 1”, it should be removed;

- Can you add a comment in the discussion section on if you think that ranitidine effectively has enough potential to be used as an anxiolytic medication, when compared to medications that are already on the market? Also are there alternative administration routes that could be possible for this drug, that would not have the disadvantages of intravenous administration?

Author Response

Reviewer-1

The current manuscript presents a quite interesting study on the anxiolytic mechanism of ranitidine. It is well done and presents relevant results. Nevertheless, some improvements should be made:

Response: We express our sincere thanks to reviewer 1 for the overall positive statements and the insightful feedback to improve the manuscript. We have carefully incorporated all the suggestions and improved the content of the manuscript.

  1. The abstract should not have abbreviations, hence they should be removed; also, the abstract lacks a final sentence, that corresponds to the study’s general conclusion(s);

Response: We thank reviewer 1 for the valid suggestion. We have removed the abbreviations in the abstract and we have appended the conclusion in the abstract.  

  1. A schematic representation should be done in the introduction section, regarding the pathophysiological aspects of the connection between histamine and its effects on the gastrointestinal tract and brain (easier for the reader to visualize);

Response: We agree with the suggestion of reviewer 1 and we have included a schematic representation to recapitulate the role of histamine in the gastrointestinal tract and brain during normal and pathophysiological conditions.

  1. A representation of the different pharmacodynamic in vivo tests, such as the open field test, light and dark box test and elevated plus maze, should be done in the form a figure, for better reader comprehension;

Response: We agree with the reviewers and we have incorporated a figure for the methodology to depict the design of behavioural test apparatuses used in the study.

  1. The figures should be provided in higher resolution; also, they should not have a title imbedded, such as “Fig 1”, it should be removed;

Response: We agree with the opinion of reviewer 1. We have revised and updated the figures as per the suggestions. In addition, we appended figures with higher resolution separately.

  1. Can you add a comment in the discussion section on if you think that ranitidine effectively has enough potential to be used as an anxiolytic medication, when compared to medications that are already on the market? Also are there alternative administration routes that could be possible for this drug, that would not have the disadvantages of intravenous administration?

Response: We thank reviewer 1 for the advice. We have incorporated a segment of text for the efficacy of ranitidine in comparison to other anxiolytic drugs and its alternative mode of administration and possible disadvantages in the discussion section.

Reviewer 2 Report

This is a very interesting study. I have only a few comments to improve the quality of the study.

Introduction

: Reference 28-31 is too old. Please, cite more recent studies and add the description.

: Lines 68-69 do not have enough evidence in this paragraph. Please, add more descriptions to support these sentences. 

Method:

What prior evidence or preliminary lab data was used to determine experimental doses of ranitidine? Please elaborate on these points.

Result:

: Figures 4E and F look similar. Could you change this?

 Also, figure 5C has a similar issue.

Conclusion

: The authors should briefly describe the main findings of the study and their implications. It contains too many unnecessary comments.

Author Response

Response to Reviewer 2

This is a very interesting study. I have only a few comments to improve the quality of the study.

Response: We extend our sincere thanks to reviewer 2 for the encouragement and feedback. The critical comments of reviewer 2 are highly helpful. We have revised the manuscript as per the suggestions from reviewer 2.

  1. Introduction

Reference 28-31 is too old. Please, cite more recent studies and add the description.

Response: We agree with reviewer 2. Additionally, we cited recent reports and updated the reference part accordingly.

  1. Lines 68-69 do not have enough evidence in this paragraph. Please, add more descriptions to support these sentences. 

Response: We thank reviewer 2 for the insightful remark. We have incorporated additional descriptions with supporting evidence as per suggestion.

  1. Method:

What prior evidence or preliminary lab data was used to determine experimental doses of ranitidine? Please elaborate on these points.

Response: We agree with the opinion of reviewer 2. Based on the available literature, we arrived at a possible dose of ranitidine as per the discussion made in the animal ethical committee meeting as a limited number of animals are recommended for experiments. We have included a description with the supporting reports for the same in the methodology portion.

  1. Result

Figures 4E and F look similar. Could you change this?

Response: We agree with the opinion of reviewer 2. We have changed the figures as per reviewer 2 suggestion.

 5.Also, figure 5C has a similar issue.

Response: We agree with the opinion of reviewer 2. We have changed the figures as per reviewer 2 suggestion.

  1. Conclusion

The authors should briefly describe the main findings of the study and their implications. It contains too many unnecessary comments.

Response: We thank reviewer 2 for the critical comment.  We have revised and updated the conclusion part.